# CD24 Gene Expression as a Risk Factor for Non-Alcoholic Fatty Liver Disease

**DOI:** 10.3390/diagnostics13050984

**Published:** 2023-03-04

**Authors:** Mona A. Amin, Halla M. Ragab, Nabila Abd El Maksoud, Wafaa Abd Elaziz

**Affiliations:** 1Department of Internal Medicine-Hepato-Gastroenterology, Kasr Al-Ainy, Cairo University, Cairo 42403, Egypt; 2Department of Biochemistry, Biotechnology Research Institute, National Research Centre, Dokki, Giza 12622, Egypt

**Keywords:** NAFLD, gene expression, CD24

## Abstract

In light of increasing NAFLD prevalence, early detection and diagnosis are needed for decision-making in clinical practice and could be helpful in the management of patients with NAFLD. The goal of this study was to evaluate the diagnostic accuracy of CD24 gene expression as a non-invasive tool to detect hepatic steatosis for diagnosis of NAFLD at early stage. These findings will aid in the creation of a viable diagnostic approach. Methods: This study enrolled eighty individuals divided into two groups; a study group included forty cases with bright liver and a group of healthy subjects with normal liver. Steatosis was quantified by CAP. Fibrosis assessment was performed by FIB-4, NFS, Fast-score, and Fibroscan. Liver enzymes, lipid profile, and CBC were evaluated. Utilizing RNA extracted from whole blood, the CD24 gene expression was detected using real-time PCR technique. Results: It was detected that expression of CD24 was significantly higher in patients with NAFLD than healthy controls. The median fold change was 6.56 higher in NAFLD cases compared to control subjects. Additionally, CD24 expression was higher in cases with fibrosis stage F1 compared to those with fibrosis stage F0, as the mean expression level of CD24 was 7.19 in F0 cases as compared to 8.65 in F1 patients but without significant difference (*p* = 0.588). ROC curve analysis showed that CD24 ∆CT had significant diagnostic accuracy in the diagnosis of NAFLD (*p* = 0.034). The optimum cutoff for CD24 was 1.83 for distinguishing patients with NAFLD from healthy control with sensitivity 55% and specificity 74.4%; and an area under the ROC curve (AUROC) of 0.638 (95% CI: 0.514–0.763) was determined. Conclusion: In the present study, CD24 gene expression was up-regulated in fatty liver. Further studies are required to confer its diagnostic and prognostic value in the detection of NAFLD, clarify its role in the progression of hepatocyte steatosis, and to elucidate the mechanism of this biomarker in the progression of disease.

## 1. Introduction

NAFLD is a clinico-pathologic syndrome that encompasses various medical entities, including simple fatty liver or simple steatosis, nonalcoholic steatohepatitis (NASH), cirrhosis, and its complications [1]. NAFLD now affects up to 25% of people around the world. The highest prevalence rate is in the Middle East (32%), followed by South America (30%), while the lowest is in Africa (13%). It also accounts for 2% of total deaths [2]. The increase in NAFLD prevalence parallels the rise in obesity and is tightly associated with metabolic comorbidities (diabetes, hypertension, insulin resistance, and dyslipidemia). It also places patients at higher risk for progressive liver disease [3].

It became clear that, as with different complex multisystem disorders, NAFLD is triggered by a variety of underlying mechanisms; the most important one of them is the alterations in hepatic and extra-hepatic lipid metabolism [4]. The study of genetic factors in NAFLD is a rapidly growing field, as they determine not only the response of different individuals to excess caloric consumption, but also the resulting metabolic derangements [5].

Cluster of differentiation 24 (CD24) is a glycophosphatidylinositol (GPI)-anchored mucin-like cell surface glycoprotein, encoded by a gene located on chromosome 6. It is expressed on mature granulocytes and B cells and regulates growth and differentiation signals to these cells. Accumulating evidence showed that abnormal over-expression of this protein is a prognostic factor in many types of cancers, resulting in cancer cell growth, proliferation, and metastasis [6]. The expression of the cell surface molecule CD24 has previously been shown to identify a subset of adipocyte progenitor cells that is crucial for the reconstitution of white adipose tissue (WAT) function in vivo, as well as a particular regulator of adipogenesis in vitro [7]. Recently, CD24 has been identified as a possible biomarker for distinguishing NAFLD/NASH. It was concluded that the mRNA expression of CD24 is upregulated in the fatty liver [8]. Additionally, Feng et al., (2021) detected that CD24 was positively associated with NAFLD severity, and it could also differentiate mild NAFLD patients from severe NAFLD patients [9]. 

Therefore, the present study aimed to identify the association between gene expression of CD24 and early stage of NAFLD. 

## 2. Subjects and Methods

The present study is a prospective study that was carried out on 80 subjects who attended outpatient clinics of the Internal Medicine Department of Kasr Al Ainy Hospital Cairo, Egypt during the period from May 2019 to December 2020 either for general health checks or to identify and treat the complications of other metabolic disorders such as diabetes or obesity.

The selected subjects were divided into two groups according to the sonographic findings of steatosis: 40 NAFLD patients with bright liver echogenicity and 40 healthy subjects with normal liver echogenicity. All cases have age ranging between 19 to 56 years old.

Those with clinical, biochemical, or histological evidence of cirrhosis, those with known causes of liver disease [viral hepatitis B and C, autoimmune hepatitis, primary biliary cirrhosis, haemochromatosis or Wilson disease], those with history of current or past excessive alcohol drinking as defined by an average daily consumption of more than 20 g alcohol, drug-induced liver disease, pregnant women and patients on hormonal contraceptive drugs (oral, parenteral), hormone replacement therapy were excluded from the study. The study was approved by Medical Research Ethical Committee of the National Research Center, Cairo, Egypt (Approval No.19-001), and informed consent was obtained from all patients.

All patients were evaluated by history and clinical examination and measurement of anthropometric parameters, such as weight (kg), height (m), body mass index (BMI; kg/m^2^), waist circumference (cm), and mid-arm circumference (cm).

Body mass index (BMI) was determined by dividing weight by square height (kg/m^2^). BMI is calculated as weight in kilograms divided by the height in metres squared. According to WHO, People with BMI = 18.5–24.9 have normal weight, people with BMI = 25.0–29.9 were classified overweight, while people with BMI ≥ 30 kg/m^2^ defines obese. BMI is calculated as weight in kilograms divided by the height in metres squared. According to WHO, in adults, overweight is defined as a BMI of 25–29.9, while a BMI ≥ 30 kg/m^2^ defines obesity. 

Waist circumference (WC) was obtained from each subject by measuring at the midpoint between the lower rib margin and the iliac crest using a conventional tape graduated in centimeters (cm). Mid-arm circumference was measured as the right upper arm measured at the midpoint between the tip of the shoulder and the tip of the elbow (olecranon process and the acromium).

Cases were divided according to their previous diagnosis or levels of fasting blood sugar: a fasting blood sugar level less than 115 mg/dL is considered normal or prediabetes. While, if the fasting blood sugar level is 126 mg/dL or higher, the patient was diagnosed diabetic.

Complete blood count was determined using the automated hematology analyzer SF-300 (Sysmex Corporation, Japan). Additionally, liver enzymes (ALT, AST, ALP, GGT), serum albumin, prothrombin time, INR, serum creatinine, lipid profile, and fasting blood sugar were measured to all individuals according to the manufacture instructions. The reagents were purchased from Spectrum Company, Cairo, Egypt. 

NAFLD fibrosis score (NFS), FIB-4, and Fast score were calculated as mentioned previously by Angulo et al. (2007) and Calès et al. (2009) [10,11] to assess fibrosis of the NAFLD patients’ group.

NFS score = −1.675 + 0.037 × age [y] + 0.094× BMI [kg/m^2^] + 1.13 × IFG/diabetes [yes = 1, no = 0] + 0.99 × AST/ALT ratio − 0.013 × platelet count [×10^9^/L] − 0.66 × albumin [g/dL]

FIB-4 score = Age [y] × AST [U/L]/platelet [×10^9^/L] × ALT [U/L]

FAST score was calculated according to Newsome et al., (2020) [12] as: FAST = {exp (–1.65 + 1.07 × ln (LSM) + 2.66 × 10^–8^ × CAP^3^ – 63.3 × AST^–1^)}/{1 + exp (–1.65 + 1.07 × ln (LSM) + 2.66 × 10^–8^ × CAP^3^ – 63.3 × AST^–1^)}(1)

Abdominal ultrasonography was performed to all individuals using the 3.5 MHz probe of Logic 6 of a General Electric machine.

### 2.1. Liver Stiffness Measurement (LSM) and Controlled Attenuation Parameter (CAP) 

Fibroscan (M probe, Echosens, Paris) was carried out by an experienced examiner in all patients (with at least 6 h of fasting) in left lateral position and the median liver stiffness of the 10 successful measurements fulfilling the criteria (success rate of greater than 60% and interquartile range/median ratio of <30%) were noted (in kPa).

The final CAP value, which ranges from 100 to 400 (dB/m), is the median of individual measurements. As an indicator of variability, the ratio of the IQR of CAP values to the median (IQR/MCAP) was calculated. The operator was blinded to the patients’ clinical data. According to the manufacturer’s instructions, in addition to previous studies, the stages of fibrosis (F0: 1–6, F1: 6.1–7, F2: 7–9, F3: 9.1–10.3, and F4: ≥10.4) were defined in kPa [13]. Moreover, steatosis stages (S0: <215, S1: 216– 252, S2: 253–296, S3: >296) were defined in dB/m [13].

### 2.2. Sample Collection

10 mL venous blood were drawn from all study participants in the morning after a 12 h fast; a portion of the blood was collected on EDTA tube for the extraction of RNA and for the determination of routine blood pictures (CBC) by Sysmex, the automated hematology analyzer SF-300, which was produced by Sysmex Corporation, Japan. The other portion was left to clot at room temperature. Serum was separated by centrifuging for 10 min at 3000 rpm. Sera were used immediately for other biochemical investigations including aspartate aminotransferase (AST), alanine aminotransferase (ALT), bilirubin, serum albumin, fasting blood glucose, cholesterol, triglycerides, HDL-C, and LDL-C according to the manufacturer’s instructions. The reagents were purchased from Spectrum Company, Cairo, Egypt.

### 2.3. CD24 Gene Expression by Quantitative Real Time-PCR (qRT-PCR):

Total RNA was isolated from whole blood using GeneJET Whole Blood RNA Purification Mini Kit (Thermo Scientific, Lithuania) following the manufacturer’s suggestions.

### 2.4. Reverse Transcription for cDNA Synthesis and Quantitative Real-Time PCR (RTqPCR)

Reverse transcription (RT) was performed to obtain cDNA from 400 ng of purified RNA using the High-Capacity cDNA Reverse Transcription Kits (Applied Biosystem, Lithuania) with random hexamers according to the manufacturer’s suggestions. A value of 10 µL of the 2X-RT master mix was pipetted into each tube and then 10 µL of RNA sample was added to it and mixed well. The tubes were centrifuged to spin down the content and to eliminate any air bubbles. After that, the tubes were placed on the PCR machine (Cleaver Scientific, UK) programmed as follows: 25 °C, 10 min, 37 °C, 120 min, and 85 °C, 5 min. After detection of cDNA concentration and purity, they were stored in −20 °C until carryover quantitative real-time PCR (QRT-PCR). 

CD-24 gene expression for enrolled samples was quantified using PowerUp SYBR Green master mix (2×) (ThermoFisher Scientific, Lithuania). The sequences for used primers were as follows:
**Primer****Primer Sequence**CD24 Forward primer5′-ACC CAC GCA GAT TTA TTC CA-3′CD24 Reverse primer5′-ACC ACG AAG AGA CTG GCT GT-3′*β*-actin Forward primer5′-TGA GCG CGG CTA CAG CTT-3′*β*-actin Reverse primer5′-TCC TTA ATG TCA CGC ACG ATT T-3′

PCR amplification was carried out in 20 μL reaction volume containing 1 µL cDNA, 10 µL PowerUp SYBR Green master mix, 7 μL nuclease-free water, and 1 µL of gene-specific forward and reverse primers as mentioned in table. The reaction was run in the Rotor-Gene Q instrument, (QIAGEN). 

Fluorescence measurements were made in every cycle, and the thermal profile was used as the follows: The amplification program included a UDG activation at 50 °C with a 2-min hold, and a dual-lock DNA polymerase at 94 °C with a 3-min hold, followed by 45 cycles with denaturation at 94 °C for 30-s, annealing at 55 °C for 30-s, and extension at 72 °C for 30-s. The expression levels of CD-24 in tested samples were expressed in the form of ∆∆CT (cycle threshold) value, which was calculated based on threshold cycle (Ct) values, corrected by β-actin expression, with the following equation.

The relative amount of CD-24 = 2^–ΔΔCt^; ΔΔCt = [ΔCt of cases − ΔCt of control]; [ΔCt = Ct (CD-24) − Ct (*β*-actin)]. The following primers were used in the quantitative real-time PCR analyses.

### 2.5. Statistical Analysis

SPSS version 16.0 (SPSS Inc., Chicago, IL, USA) was used for statistical analysis with a two-side significant criterion at *p* < 0.05. The clinical data were expressed as mean ± SD (continuous, normally distributed variables). Categorical data were summarized as percentages. The significance for the difference between groups was determined by using a two-tailed Student’s t-test. Additionally, qualitative variables were assessed by chi-squared χ^2^-test. Correlations between different parameters were performed using Pearson’s and spearman’s correlation coefficients. A receiver operating characteristic (ROC) curve was plotted to assess the diagnostic power of CD24 in NAFLD and controls, and the area under the curve (AUC) greater than 0.5 considered to be statistically significant. The probability (*p*) values of ≤0.05 were considered statistically significant and indicated, while *p* > 0.05 was considered statistically not significant and indicated NS.

## 3. Results 

The present study is a case-control study recruited 80 adult subjects, (28 males and 52 females). Their age ranged from 19 to 56 years. 

The demographic, anthropometric, clinical, and biochemical characteristics of both groups (NAFLD and controls) are summarized in Table 1. Patients with NAFLD were significantly older than controls (mean age 42.18 ± 11.1 4 y vs. 29.65 ± 6.63 y, *p* < 0.0001). There were more males in the control group (45%), but the majority was females in the NAFLD group (75%). NAFLD patients exhibited a higher mean BMI (31.8 ± 2.9 kg/m^2^) than the control group (23.76 ± 1.4 kg/m^2^) (*p* < 0.001).

Patients with NAFLD had a higher prevalence of hypertension and diabetes mellitus in comparison to healthy controls (*p* < 0.001) (Table 1).

Among studied NAFLD patients, 22.5% had a family history of diabetes, and 30% had family history of liver disease, and 62.5% of NAFLD cases (*n* = 25) have enlarged liver size on ultrasound. 

The mean serum fasting blood glucose was significantly higher in NAFLD patients than that in controls (122.6 ± 40.97 vs. 96.03 ± 7.77); (*p* < 0.001). In addition, hemoglobin levels were lower in NAFLD cases (11.56 ± 1.4 (g/dL) than in healthy controls (12.81 ± 1.06 (g/dL), (*p* < 0.001). No significant difference was observed in total leucocytic count (TLC) and platelet count between the NAFLD and control groups (*p* > 0.05).

NAFLD patients had significantly higher serum levels of aspartate aminotransferase (AST), alanine aminotransferase (ALT), alkaline phosphatase (ALP), and gamma-glutamyl transferase (GGT) compared to healthy controls (*p* < 0.001). On the other hand, the mean albumin level was almost normal (3.8 ± 0.38 g/dL) in the NAFLD group. 

There was a significant elevation in total cholesterol, triglycerides, and LDL-cholesterol among NAFLD patients compared to controls, while there was significant decrement in HDL in the NAFLD group as opposed to controls (*p* < 0.05). 

Table 2 shows clinical and biochemical characteristics of participants stratified by sex and presence/absence of NAFLD. In both sexes, participants with NAFLD were older, had a higher BMI, as well as a higher prevalence of diabetes. Levels of hemoglobin was significantly lower in female cases compared to male cases in NAFLD group (*p* = 0.001). However, ALT and AST levels were significantly higher in male NAFLD cases compared to female NAFLD casess (*p* = 0.009 and *p* = 0.038; respectively) (Table 2).

The mean Fibroscan value in all NAFLD patients was 5.1 ± 0.99 (kPa), indicating that all patients had mild fibrosis with a stage less than 2. Thirty patients had fibrosis belonging to stage 0, while the rest had fibrosis stage 1. Mean Fibroscan values for cases with fibrosis stages 0 and 1 were 4.7 ± 0.67 and 6.5 ± 0.3 (kPa), respectively. There was a statistically significant difference in liver stiffness measurements in patients with stage 0 fibrosis as compared to stage 1 fibrosis (*p* < 0.001). In addition, there was a stepwise increase in Cap score parallel to the increase in severity of liver fibrosis (*p* < 0.001) (Table 3). 

This study showed that both NFS and FIB-4 score were similar in patients with fibrosis stages 0 and those with fibrosis stages 1 (*p* > 0.05). This may be due to that all cases included in our study have mild fibrosis. Additionally, performances of FIB-4 and NFS to rule in advanced fibrosis are rather inadequate, meaning that further assessment with another test is needed in case of positive results.

According to the RT-PCR results, it was detected that expression of CD24 was significantly higher in patients with NAFLD than healthy controls. The median fold change in the expression of CD24 was 6.56 higher in NAFLD cases compared to control subjects (Figure 1). 

The present study showed higher expression of CD24 in female cases with NAFLD compared to male cases (fold change was 6.9 in females vs 4.4 in males, but without significant difference; *p* = 0.262) (Figure 2).

Additionally, CD24 expression was higher in cases with fibrosis stage F1 compared to those with fibrosis stage F0, as the mean expression level of CD24 was 7.19 in F0 cases as compared to 8.65 in F1 patients, but without significant difference (*p* = 0.588). Furthermore, there was no difference in CD24 fold change between overweight patients (median fold change = 9) and obese cases (median fold change = 5.89) (*p* = 0.447) (Figure 3). Additionally, the median fold change in CD24 in diabetic cases was seven compared to 5.13 in non-diabetic cases (*p* = 0.609) (Figure 4).

### 3.1. Evaluation of the Diagnostic Accuracy of CD24 Gene Expression for Distinguishing Patients with NAFLD from Healthy Controls

Figure 5 illustrates the ROC plots to assess the diagnostic accuracy of CD24 ∆CT to distinguish patients with NAFLD from healthy controls.

ROC curve analysis showed that CD24 ∆CT had significant diagnostic accuracy in the diagnosis of NAFLD (*p* = 0.034). ROC curve showed the optimum cutoff for CD24 was 1.83 for distinguishing patients with NAFLD from healthy control with sensitivity 55% and specificity 74.4%; and an area under the ROC curve (AUROC) 0.638 (95% CI: 0.514–0.763).

### 3.2. Correlation between Different Non-Invasive Fibrosis Markers and CD24 Gene Expression

Table 4 shows the correlation of Kpa, CAP, FAST, NFS, and FIB-4 with CD 24 gene expression. Pearson’s correlation test showed positive significant correlation between CD24 and NFS (r = 0.356, *p* = 0.001). 

By binary logistic regression analysis, none of the examined parameters found to be significant determinant of NAFLD after adjusting the effects of potential cofounders of age, gender, suffering of diabetes, and BMI, respectively (Table 5).

## 4. Discussion

NAFLD is known nowadays as the most common liver disorder in the 21st century. It is diagnosed by the presence of more than 5% fat accumulation in liver cells without excess alcohol consumption or secondary causes of fat accumulation in the background. Approximately 25% of the world’s adult’s population has NAFLD, and the prevalence is still increasing [13]. 

NAFLD may eventually deteriorate to HCC as a result of excessive lipid accumulation, liver cell damage, immune system dysfunction, which leads to scarring, and permanent liver damage [14]. In light of increasing NAFLD prevalence, early detection and diagnosis are needed for decision-making in clinical practice and could be helpful in the management of patients with NAFLD.

The present study showed a significant trend of elder age with the progression of non-alcoholic fatty liver disease. This finding substantiates previous findings in the literature, which suggested that the prevalence of NAFLD increases with increasing age [15]. 

The present study showed that, regarding gender distribution, there were more males in the control group (45%) compared to the NAFLD group (25%), but the majority was females in the NAFLD group (75%). These results revealed that there was no statistically significant difference between both studied groups according to gender as *p* = 0.061. The explanation for the gender difference is different distributions of fat mass by gender, e.g., more abdominal visceral adipose tissue in male and more subcutaneous adipose tissue mass in female. Additionally, previous results showed that Hispanic women having a higher risk for NAFLD compared to men, whereas, for the non-Hispanic population, the prevalence of NAFLD is more frequent in males [16]. Additionally, Lonardo et al. mentioned that gender is one of the main cause of variation in NAFLD risk factors. They also detected that NAFLD is more common and more severe in men than women. However, it is more common in women after menopause, indicating that estrogen may be beneficial [17].

In the current study, the incidence of NAFLD has been increasing in concert with the presence of multiple metabolic disorders, such as dyslipidemia, diabetes, hypertension, and visceral obesity. As expected, the incidence of diabetes and hypertension was significantly higher in patients suffering from NAFLD. This is in good agreement with previous studies that mentioned impaired glucose tolerance as an independent risk factor for the progression of NAFLD [18,19]. 

According to the International Diabetes Federation (IDF), the prevalence of DM among Egyptian adults is 15.2%, which may be an underestimation [20]. Lonardo et al. reported that patients with T2DM had 80% higher liver fat contents compared to non-diabetic patients [21].

Additionally, Lee, et al., (2019), mentioned that compared to the general population (around 25%), 50% to 70% of people with diabetes have NAFLD, and NAFLD severity (including fibrosis) tends to be worsened by the presence of diabetes [22]. 

Additionally, another study carried out on the Egyptian college students showed that around 1 in 3 had steatosis, and 1 in 20 had fibrosis. The prevalence of NAFLD in young adults was estimated to be 31.6%, which is perfectly in line with the 31.8% prevalence rate found in a meta-analysis of numerous epidemiological studies across general Middle Eastern populations. It is known that the Middle East and North Africa region has one of the highest prevalence rates of NAFLD globally, and that Egypt ranked among the highest 10 nations with obesity prevalence. Combing both may explain our unexpected observation. In our cohort, 59 (49.2%) of participants were overweight or obese [23]. 

NAFLD is caused by a variety of different molecular pathways and cellular alterations. The molecular pathways of NAFLD pathogenesis in the liver have been identified in several studies. The major genes linked to illness development and the underlying functional pathways are yet unknown, and whether the differentially expressed CD24 is involved in hepatic lipid metabolism is still unclear. 

Microarray technologies have revealed a large number of new molecular markers (DNA, RNA, and protein) in recent years. Further research is needed to confirm the clinical utility of these impending novel indicators in relation to hepatic steatosis. CD24 is one of these markers, which was recently reported by Huang et al. as a possible biomarker in the course of hepatocyte steatosis [8]. 

Various studies have recently discovered that CD24 expression is relatively high in many human malignancies, including HCC [24,25,26,27,28]. Additionally, CD24 overexpression has been correlated with increased invasiveness, proliferation, and metastasis [29]. It was previously identified that a subpopulation of adipocyte progenitor cells with the expression of the cell surface molecule CD24 being necessary for reconstitution of white adipose tissue function in vivo as well as being a key regulator of adipogenesis in vitro [30].

In our study, we investigated the association between CD24 gene expression and the prevalence of NAFLD.

The current study found that CD24 gene expression was considerably greater in NAFLD cases compared to controls, and the normalized CD24 gene expression in NAFLD was up-regulated 6.56-fold. These findings suggest that the CD24 gene is important in the development of NAFLD. This could be related to CD24 gene expression’s impact on the immune/inflammatory response via T-cell activation [31]. Several immune cell-mediated inflammatory processes are involved in NAFLD and its progression to NASH. They also influence the generation of cytokines by necrotic liver cells [32]. 

This confirms the previous results detected by Feng et al., who observed the up-regulation of CD24 gene expression in the livers of HFD-induced NAFLD mice and in cultured HepG2 cells exposed to glucolipotoxicity (palmitic acid or advanced glycation end products) [9]. Up until now, the precise role and the underlying mechanisms of CD24 in NAFLD progression remain unclear. However, Huang and his colleague identified the prominent correlation between CD24 and NAFLD/NASH. They mentioned that CD24 could play a key role in one of the pathways that may cause IR and may induce NAFLD/NASH in humans including [“glycolysis/gluconeogenesis”, “p53 signaling pathway” and “glycine”, serine and threonine metabolism [8].

Additionally, CD24 expression was higher in cases with fibrosis stage F1 compared to those with fibrosis stage F0, as the mean expression level of CD24 was 7.19 in F0 cases as compared to 8.65 in F1 patients, but without significant difference (*p* = 0.588). This may be because that all cases included in the present study have mild fibrosis. This results most be confirmed by other studies based on large number of samples and on patients with severe stage of fibrosis. 

The changes in liver tissue-transcriptome in a subset of 25 mild-NAFLD and 20 NASH biopsies were examined in a cross-sectional study. CD24 was revealed to be one of five differentially expressed genes (DEGs) positively linked with disease severity and to be main classifiers of mild and severe NAFLD [33].

Additionally, CD24-positive cells isolated from hepatocellular carcinoma cell lines exhibited stemness properties, such as self-renewal, chemotherapy resistance, metastasis, and tumorigenicity [34]. These results indicate that CD24 may play a role in hepatocyte injury and promote regeneration during the development and progression of NAFLD. 

Another Egyptian study detected that CD24 polymorphism 170 CT/TT may affect the incidence of infection with CHC, as well as HCC [35]. They revealed that the P170T allele, which is expressed at a higher level than P170C, encodes a certain protein, which is responsible for the progression of chronic HCV infection by affecting the efficiency of cleavage of posttranslational GPI. Additionally, Robert and Pelletier (2018) showed that the P170T allele affects the progression of chronic HCV infection through posttranslational mechanisms [36]. Another study by Kristiansen et al. (2010) also suggested that CD24 SNPs are prognostic markers for hepatic carcinoma [37]. 

Interestingly, CD24 was also up-regulated in the NAFLD patients with type 2 diabetes than its expression in non-diabetic cases, but without significant difference. 

Another study carried out by Shapira et al. (2021) reported that CD24 may negatively regulate peroxisome proliferator-activated receptor gamma (PPAR-γ) expression in male mice. This gene is a regulator of adipogenesis that plays a role in insulin sensitivity, lipid metabolism, and adipokine expression in adipocytes. Furthermore, they concluded the association between the CD24 and insulin sensitivity, suggesting its possible mechanism for diabetes [38].

## 5. Conclusions

The current study found CD24 gene expression was considerably greater in NAFLD cases compared to controls. This could indicate that CD24 may contribute to hepatic steatosis, but a current study showed that it cannot be used as an independent predictor of NAFLD.

Further studies are required to confer its diagnostic and prognostic value in the detection of NAFLD, as well as to clarify its role in the progression of hepatocyte steatosis in patients with advanced stage of fibrosis and to elucidate the mechanism of this biomarker in the progression of disease. However, our study is limited because of the small sample size, because all participants in this study have early stage of NAFLD, and because accurate diagnosis of liver fibrosis or hepatocellular injury are invasive and very expensive. Although abdominal ultrasonography has low sensitivity for detecting mild-NAFLD as reported in the previous literature, it is the best low-cost available non-invasive technique to detect NAFLD. Because of ethical considerations, we did not rely on the liver biopsy for diagnosis, as none of the patients had clinical manifestations. Moreover, the studied patients considered themselves healthy and refused to undergo further invasive investigations, including pathological examinations via liver biopsy to detect fibrosis.

## Figures and Tables

**Figure 1 diagnostics-13-00984-f001:**
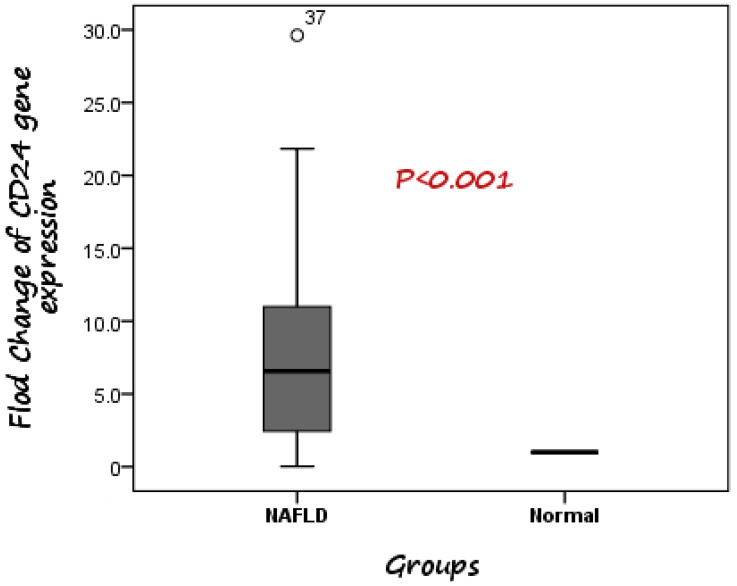
Comparison between CD24 gene expression in both NAFLD and control. The median fold change was 6.56 higher in NAFLD cases compared to control subjects.

**Figure 2 diagnostics-13-00984-f002:**
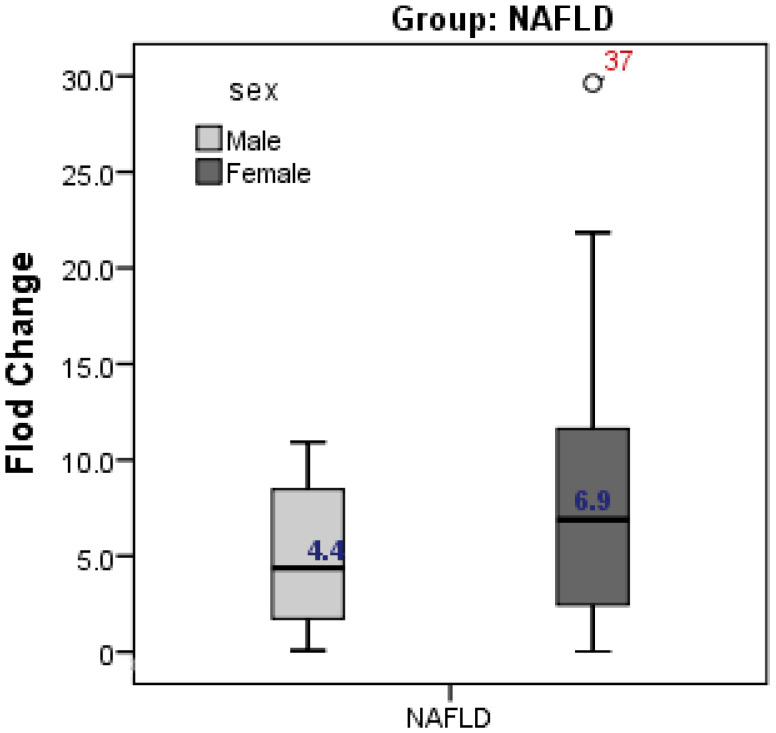
Gender difference in CD24 gene expression in relation to NAFLD. Expression of CD24 was higher in female than male but without significant difference.

**Figure 3 diagnostics-13-00984-f003:**
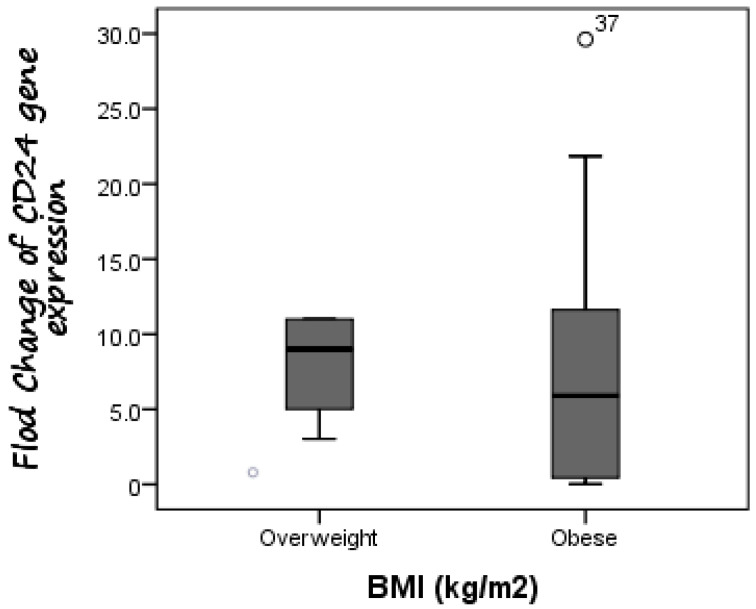
Comparison between CD24 gene expression in both overweight and obese NAFLD patients. There was no difference in CD24 fold change between overweight patients (median fold change = 9) and obese cases (median fold change = 5.89).

**Figure 4 diagnostics-13-00984-f004:**
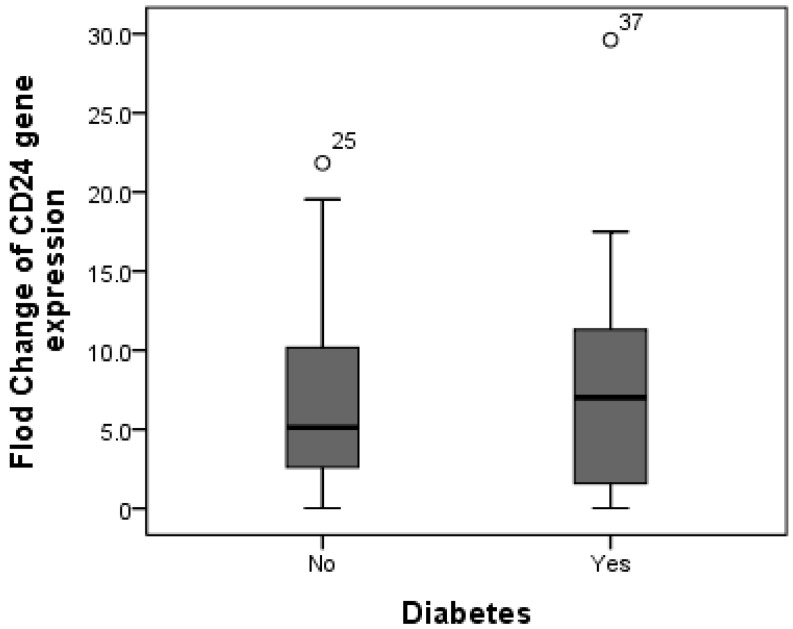
Comparison between median CD24 gene expression in both non-diabetic and diabetic NAFLD patients. The median fold change in CD24 in diabetic cases was seven compared to 5.13 in non-diabetic cases.

**Figure 5 diagnostics-13-00984-f005:**
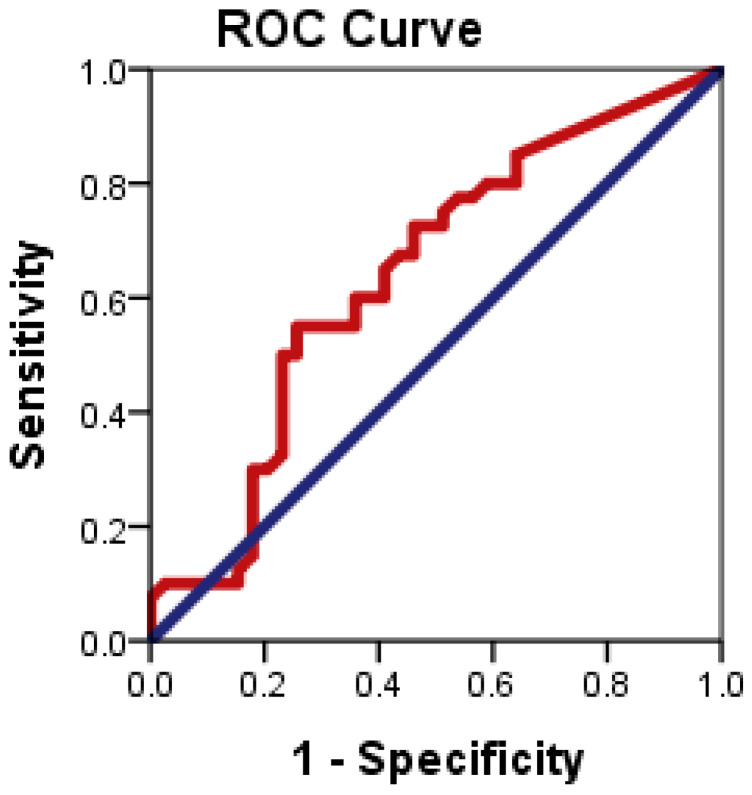
ROC curve of CD24 gene expression to differentiate between NAFLD cases from health controls.

**Table 1 diagnostics-13-00984-t001:** Characteristics of Study Participants According to NAFLD Status.

Variable		NAFLD Group (N = 40)	Control Group (N = 40)	*p*-Value
	Groups			
Age (Yrs.)		42.18 ± 11.14	29.65 ± 6.63	<0.001 **
Gender ^a^				0.061
Male/Female		10/30	18/22	
Percentage of Male		(25%)	(45%)	
Hypertension ^a^				0.006 **
Absent		33 (82.5%)	40 (100%)	
Present		7 (17.5%)	0 (0%)	
Diabetes ^a^				<0.001 **
Absent		20 (50%)	40 (100%)	
Oral hypoglycemic		9 (22.5%)	0 (0%)	
on insulin		11 (27.5%)	0 (0%)	
Family history of diabetes ^a^				1.000
No		31 (77.5%)	31 (77.5%)	
Yes		9 (22.5%)	9 (22.5%)	
Family history of liver diseases ^a^				0.805
No		28 (70%)	29 (72.5%)	
Yes		12 (30%)	11 (27.5%)	
BMI (kg/m^2^)		32.42 ± 3.59	23.66 ± 1.3	<0.001 **
Waist circumference (cm)		117.9 ± 9.4	74.3 ± 6.8	<0.001 **
Mid-arm circumference (cm)		31.8 ± 4.3	25.5 ± 1.5	<0.001 **
Ultrasound finding ^a^				
Enlarged liver size		25 (62.5%)	0 (0%)	<0.001 **
Laboratory variables				
Fasting blood glucose (mg/dL)		122.6 ± 40.97	96.03 ± 7.77	<0.001 **
HB (g/dL)		11.56 ± 1.4	12.81 ± 1.06	<0.001 **
Platelets count (10^3^/µL)		236 ± 72.78	260.63 ± 56.79	0.096
Total leucocytic count (10^3^/µL)		6.79 ± 1.91	7.24 ± 1.7	0.273
ALT (U/L)		47.9 ± 18.3	30.6± 4.9	<0.001 **
AST (U/L)		33.55 ± 14.9	23.6 ± 4.6	<0.001 **
Total bilirubin (mg/dL)		0.76 ± 0.24	0.68 ± 0.2	0.132
ALP (U/L)		155.7± 37.5	115.02± 15.2	<0.001 **
GGT (U/L)		91.1± 64.2	34.5± 9.2	<0.001 **
Total Protein (g/dL)		8.1± 0.2	7.9± 0.24	0.007 **
INR		1.05 ± 0.1	1.0 ± 0.0	0.003 **
Serum Albumin (g/dL)		3.8 ± 0.38	3.8 ± 0.3	0.939
Cholesterol (mg/dL)		150.88 ± 32.4	101.65 ± 19.76	<0.001 **
Triglycerides (mg/dL)		168.5 ± 44.15	143.9 ± 29.7	0.005 **
LDL (mg/dL)		133.7 ± 34.6	104.02 ± 16.1	<0.001 **
HDL (mg/dL)		48.4 ± 15.5	55 ± 12.7	0.04 *

BMI, body mass index; Hb, hemoglobin; ALT, alanine aminotransferase; AST, aspartate aminotransferase; GGT, gamma-glutamyl transferase; ALP, alkaline phosphatase; HDL, high-density lipoproteins; LDL, low-density lipoproteins. All data are displayed as mean and standard deviation. (SD) unless otherwise indicated. ^a^ Categorical data were summarized as percentages. Differences were analyzed with χ^2^ (chi square) tests and Fisher’s exact test. * = significant *p* value (≤0.05), ** = highly significant *p* value (≤0.01).

**Table 2 diagnostics-13-00984-t002:** Clinical and biochemical characteristics of participants stratified by sex and presence/absence of NAFLD.

Variable		NAFLD Group (N = 40)		*p*-Value	Control Group (N = 40)		*p*-Value
	Groups	Male N = 10	Female N = 30		Male N = 18	Female N = 22	
Age (Yrs.)		44.8 ± 9.1	41.3 ± 11.7	0.396	28.5 ± 4.7	30.59 ± 7.8	0.305
Hypertension				0.338			-
Absent		7 (70%)	26 (86.7%)		18 (100%)	22 (100%)	
Present		3 (30%)	4 (13.3%)		0 (0%)	0 (0%)	
Diabetes				0.322			-
Absent		7 (70%)	13 (43.3%)		18 (100%)	22 (100%)	
Oral hypoglycemic		1 (10%)	8 (26.7%)		0 (0%)	0 (0%)	
on insulin		2 (20%)	9 (30%)		0 (0%)	0 (0%)	
Family history of diabetes				0.827			0.970
No		8 (80%)	23 (76.7%)		14 (77.8%)	17 (77.3%)	
Yes		2 (20%)	7 (23.3%)		4 (22.2%)	5 (22.7%)	
Family history of liver diseases				0.231			0.173
No		9 (90%)	19 (63.3%)		11 (61.1%)	18 (81.8%)	
Yes		1 (10%)	11 (36.7%)		7 (38.9%)	4 (18.2%)	
BMI (kg/m^2^)		32.51 ± 3.1	32.39 ± 3.8	0.929	23.95 ± 1.1	23.4 ± 1.33	0.183
Laboratory variables							
Fasting blood glucose (mg/dL)		108.7 ± 38.9	127.23 ± 41.2	0.220	93.78 ± 6.3	97.86 ± 8.5	0.098
HB (g/dL)		12.72 ± 1.03	11.17 ± 1.3	0.001 **	12.9 ± 0.99	12.7 ± 1.1	0.572
Platelets count (10^3^/µL)		252.3 ± 73.3	230.57± 73.03	0.421	270.9 ± 64.8	252.2 ± 49.3	0.305
Total leucocytic count (10^3^/µL)		7.18 ± 1.69	7.07 ± 1.9	0.875	6.7 ± 1.2	7.3 ± 1.83	0.253
ALT (U/L)		60.7 ± 18.6	43.7 ± 16.4	0.009 *	29.9± 5.1	31 ± 4.8	0.489
AST (U/L)		44.7 ± 18.9	29.8 ± 11.4	0.038 *	22.8 ± 3.9	24.3 ± 5	0.312
Total bilirubin (mg/dL)		0.82 ± 0.25	0.74 ± 0.24	0.331	0.69 ± 0.2	0.66 ± 0.2	0.605
ALP (U/L)		156.4± 40.4	155.5 ± 37.2	0.949	117.3± 15	113.1 ± 15.4	0.390
GGT (U/L)		111.1± 61.9	84.4 ± 64.5	0.259	35.6± 10.3	33.5 ± 8.3	0.488
Total Protein (g/dL)		8.1± 0.34	8.04± 0.23	0.574	7.8± 0.24	7.9 ± 0.24	0.277
Serum Albumin (g/dL)		3.8 ± 0.37	3.8 ± 0.39	0.962	3.8 ± 0.34	3.8 ± 0.26	0.489
Cholesterol (mg/dL)		158 ± 22.4	148.5 ± 35.15	0.429	105.1 ± 22.1	98.8 ± 17.6	0.323
Triglycerides (mg/dL)		146.3 ± 52.4	175.83 ± 39.3	0.066	152.2 ± 25.8	137.14 ± 31.5	0.112
LDL (mg/dL)		128.4± 34.2	135.5 ± 35.2	0.582	105.11± 17	103.9 ± 15.7	0.811
HDL (mg/dL)		46.5 ± 20.17	49± 14.02	0.665	52.4 ± 6.5	57.09 ± 15.9	0.253

BMI, body mass index; Hb, hemoglobin; ALT, alanine aminotransferase; AST, aspartate aminotransferase; GGT, gamma-glutamyl transferase; ALP, alkaline phosphatase; HDL, high-density lipoproteins; LDL, low-density lipoproteins. * = significant *p* value (≤0.05), ** = highly significant *p* value (≤0.01).

**Table 3 diagnostics-13-00984-t003:** Results of Fibroscan, CAP, NAFLD Fibrosis Score, FIB 4, and FAST Score in NAFLD Group.

	Group I Total NAFLD Patients N = 40	Fibrosis Stage (F0) N = 30	Fibrosis Stage (F1) N = 10	*p*-Value
Fibroscan (kPa)				
Mean ± SD	5.1 ± 0.99	4.7 ± 0.67	6.5 ± 0.3	<0.001 **
Range	3.8–6.9	3.8–5.9	6.2–6.9	
CAP (dB/m)				
Mean ± SD	263.9 ± 11.61	260.4 ± 10.9	274.2 ± 6.44	<0.001 **
Range	242–286	242–286	265–286	
FAST score				
Mean ± SD	0.203 ± 0.139	0.2377 ± 0.14	0.099 ± 0.09	0.001 **
Range	0.04–0.47	0.05–0.47	0.04–0.25	
NAFLD fibrosis score				0.147
Mean ± SD	−1.37 ± 1.38	−1.19 ± 1.4	−1.92 ± 1.01	
Range	−4.1–1.22	−4.1–1.22	−3.2–0.33	
FIB 4				
Mean ± SD	0.919 ± 0.46	0.96 ± 0.5	0.81 ± 0.28	0.376
Range	0.33–2.39	0.33–2.39	0.36–1.14	

** = highly significant *p* value (≤0.01).

**Table 4 diagnostics-13-00984-t004:** Correlation between different non-invasive fibrosis markers and CD24 gene expression.

Parameters	CD24 Gene Expression	
	r	*p*-Value
Kpa (kPa)	−0.070	0.677
CAP(dB/m)	−0.050	0.764
FAST	−0.006	0.970
NFS	0.356	0.001 **
FIB-4	0.090	0.432

** = highly significant *p* value (≤0.01).

**Table 5 diagnostics-13-00984-t005:** Binary logistic regression analysis using NAFLD as the dependent variable after adjusting the major confounders.

Variables	Coefficient	Sig.	Odds Ratio
age	0.695	0.999	2.003
sex	−8.443	0.999	0.000
DM	27.061	1.000	565,692,536,831.086300
BMI	−5.426	0.997	227.262
Fold change of CD24	1.372	1.000	3.943

## Data Availability

The datasets used and/or analyzed during the current study are available from the corresponding author on reasonable request.

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
