# Peer review of "CD24 Gene Expression as a Risk Factor for Non-Alcoholic Fatty Liver Disease"

_diagnostics, 2023, doi:10.3390/diagnostics13050984_

Round 1

Reviewer 1 Report

In the present manuscript the authors report the results of a cross-sectional study investigating the abilitiy of CD24 expression to diagnose liver steatosis. The topic is certainly worth investigating given the prevalence of NAFLD in the general population. The manuscript is generally well-written and easy to follow.

I have the following comments:

1.       Please refrain from commenting the relevance of your results in the abstract section (“Interestingly”).

2.       Please better describe how patients and controls were recruited. Did you try to recruit all patients referred for liver ultrasound at your institution?

3.       The two groups are extremely different in terms of sex, age and comorbidities. I find it peculiar that 50% of patients with NAFLD and a mean age of 42 years had diabetes and 27% (more than half of those with diabetes) were treated with insulin. Again it seems that some sort of selection bias is present.

4.       Please report in how many patients Fibroscan results were unreliable (IQR/Median >30%)

5.       Given the differences between the two groups I would run a logistic regression mode using steatosis as the outcome and including age, sex, BMI, diabete and CD24 as predictors, to evaluate whether CD24 is retained as an independent predictor or not.

6.       In the discussion section I would mention differences between men and women in NAFLD and fibrosis prevalence and association with fat deposition (I suggest mentioning doi: 10.1093/ajcn/nqac059.)

Reviewer 2 Report

Recommendations

Comments to the author: First of all, congratulate the authors for their excellent work. The manuscript follows a solid methodology, is simple and has a clear objective. The conclusions fully correspond to the results obtained.

The article does not cause any concern. The manuscript did not cause any ethical problems. The statistical analysis corresponds to the study. All references to publications presented by the authors in the article are necessary and correct, made in the correct style. Of the 29 links presented in the article, 17 links from the last 5 years (2017-2022).

I still have some questions and recommendations.

minor issues

1.- It must be specified how the anthropometric parameters have been measured.

2.- It must specify how overweight and obesity are determined, it is not defined in the introduction or material and methods, and then a comparison is made.

3.- The concept of diabetic patient must be specified, it is not defined in the introduction or material and methods, and then a comparison is made.

4.- Line 67 specifies that the range of included patients is from 18 to 60 years of age. While line 164 specifies that all the people included in the sample are between 19 and 56 years old. It must be unified.

5.- The "fasting blood glucose" glucose figures in Table 1 do not agree with those in the text, lines 180 and 181.

6.- The hemoglobin figures in Table 1 do not match those in the text, lines 181 and 182.

Reviewer 3 Report

Please refer to the attachment

Author Response

answers in attached file 

Round 2

Reviewer 1 Report

I have no further comments